# Optimizing Polymer Solar Cells Using Non-Halogenated Solvent Blends

**DOI:** 10.3390/polym11030544

**Published:** 2019-03-22

**Authors:** Guler Kocak, Desta Gedefaw, Mats R. Andersson

**Affiliations:** 1Flinders Institute for Nanoscale Science and Technology, Flinders University, Sturt Road, Bedford Park, Adelaide, SA 5042, Australia; guler.kocak@flinders.edu.au (G.K.); gedefaw_d@usp.ac.fj (D.G.); 2School of Biological and Chemical Sciences, The University of South Pacific, Laucala Campus, Private mail bag, Suva Fiji

**Keywords:** OPV, non-halogenated, environmentally friendly, morphology, solvent additive

## Abstract

More environmentally friendly polymer solar cells were constructed using a conjugated polymer, poly (2,5-thiophene-alt-4,9-bis(2-hexyldecyl)-4,9-dihydrodithieno[3,2-c:3′,2′h][1,5] naphthyridine-5,10-dione, PTNT, as a donor material in combination with PC_71_BM as an acceptor in a bulk heterojunction device structure. A non-halogenated processing solvent (*o*-xylene) and solvent additives that are less harmful to the environment such as 1-methoxynaphthalene (MN) and 1-phenylnaphthalene (PN) were used throughout the study as processing solvents. The most widely used halogenated solvent additives (1,8-diiodooctane (DIO) and 1-chloronaphthalene (CN)) were also used for comparison and to understand the effect of the type of solvent additives on the photovoltaic performances. Atomic force microscopy (AFM) was employed to investigate the surface morphology of the films prepared in the presence of the various additives. The best-performing polymer solar cells provided a high open-circuit voltage of 0.9 V, an efficient fill factor of around 70%, and a highest power conversion efficiency (PCE) of over 6% with the use of the eco-friendlier *o*-xylene/MN solvent systems. Interestingly, the solvent blend which is less harmful and with low environmental impact gave a 20% rise in PCE as compared to an earlier reported device efficiency that was processed from the chlorinated solvent *o*-dichlorobenzene (*o*-DCB).

## 1. Introduction

Organic photovoltaics (OPV) technology has enabled the easy and cost-effective production of solar cells being dissimilar from their inorganic counterparts. Different fabrication approaches are still required for high efficiencies to reach the full potential of organic solar cells. For an eco-friendly approach, highly efficient inverted bulk heterojunction (BHJ) solar cells are the most prominent candidates.

In BHJ polymer solar cells, donor and acceptor moieties are blended in common organic solvents [1] such as chloroform, chlorobenzene, and *o*-dichlorobenzene (*o*-DCB) to prepare interpenetrating donor–acceptor films. Even though these solvents are known for their lack of environmental friendliness and hence are less preferable for the large-scale production of solar cells [2], they are still largely used to fabricate photovoltaic devices [3]. In addition, the use of high boiling point solvent additives [4,5,6] was generally practiced as an effective morphology optimization technique for achieving high efficiency and large-scale processing [7,8,9]. In this case, the most well-known halogenated additives are 1,8-diiodooctane (DIO) and 1-chloronaphthalene (CN), and non-halogenated ones include xylenes and trimethylbenzenes [10].

Ideal donor polymers for an organic solar cell require narrow band gaps for efficient light harvesting, good charge transport abilities, and high solubility and film-forming properties. Ubiquitous polymers were developed in the past and tested in polymer solar cells in order to find a donor polymer that fulfils the desired properties. Among these polymers, poly (2,5-thiophene-alt-4,9-bis(2-hexyldecyl)-4,9-dihydrodithieno[3,2-c:3′,2′h][1,5] naphthyridine-5,10-dione (PTNT) is a promising donor polymer that exhibits a semi-crystalline nature in the solid state. Interestingly, although it has a slightly broad band gap around 2.2 eV, it still presents good device performance owing to the enhanced solid-state packing, and as a result offers an increased charge carrier mobility [11] and efficient dissociation of charge carriers. Its highest photovoltaic device efficiency was around 5% (~200 nm thickness) when blended with PC_71_BM as reported in a previous study [7]. Fullerenes are good electron acceptors and by the addition of functional groups to the fullerene cage, the most commonly used acceptors PC_61_BM and PC_71_BM are obtained. PC_71_BM has a stronger light absorption than the PC_61_BM acceptor. The device efficiency of a solar cell strongly depends on the fine-tuning of the photoactive layer morphology. Since open circuit voltage (Voc) stems from the donor–acceptor energy level differences (HOMO level of donor and LUMO level of acceptor) [12,13], it is critical to improving the short-circuit current density (Jsc) for high efficiency. The current density of photovoltaic devices increases as the charge transport in the donor–acceptor blend improves as well as with the optimization of the donor–acceptor interfaces [12]. In the past, various techniques [13,14] such as thermal annealing, solvent vapor annealing, solvent additives, and other additives such as polymers and inorganic nanocrystals [15,16,17,18] were investigated to fine-tune the morphology of the donor–acceptor blend to improve the short-circuit current density. Solvent additives have been particularly preferred for large-area applications [19,20] due to their easy applicability [21]. Less toxic solvents and solvent additives are particularly of high interest from the point of health and environmental issues, and results have been very promising [22,23].

In this work, we focused on depositing the active layer blend from the less harmful organic solvent (*o*-xylene) and different solvent additives. Their functions on the optimization of the films prepared from a blend of PTNT and PC_71_BM were investigated. We correlated the devices’ performances with the morphology of the films. Among the different solvent additives investigated, 1-methoxynaphthalene (MN) gave a better morphology and hence an optimized J_SC_ of around 10 mAcm^−2^ and a power conversion efficiency (PCE) of ~6% were achieved. A slightly lower performance reaching a maximum of 5.4% PCE was recorded with the use of 1-phenylnaphthalene (PN) as a non-halogenated solvent additive. Moreover, the highest-performing solvent additives are non-toxic to the environment and hence suitable for large-scale production. Note that chlorinated solvent additives (DIO and CN) were also used for device fabrication as a comparison. Structures of the PTNT polymer, processing solvent, and additives used in this study are given below in Figure 1. To the best of our knowledge, 1-methoxynaphthalene (MN) has not previously been used to improve the morphology of polymer solar cells.

## 2. Materials and Methods 

### 2.1. Materials 

PC_71_BM (purity > 99%) was purchased from Solenne BV (Groningen, The Netherlands). Poly(2,5-thiophene-alt-4,9-bis(2-hexyldecyl)-4,9-dihydrodithieno[3,2-c:3′,2′-h][1,5]naphthyridine-5,10-dione) (PTNT) was synthesized as described in a previous publication [7]. The molecular weight of PTNT was *M*_n_ = 55.7 kg/mol and *M*_w_ = 163.2 kg/mol relative to polystyrene standards, using an Agilent PL-GPC 220 Integrated High Temperature GPC System with refractive index detectors using 3 × PLgel 10 µm MIXED-B LS, 300 × 7.5 mm^2^ columns with 1,2,4-trichlorobenzene at 150 °C as the eluent. The polymer was blended with PC_71_BM in a 2:3 weight ratio. *o*-Xylene, DIO, MN, PN, and CN were purchased from Sigma Aldrich Co. LLC. (NSW, Australia) and used without further purification. Ag and MoO_3_ were also purchased from Sigma Aldrich Co. LLC. (NSW, Australia).

### 2.2. Substrate Preparation

Patterned ITO (indium tin oxide) glass substrates with a sheet resistance of (10 Ω/sq., Xin Yan Technology Ltd., Kwun Tong, Kowloon, Hong Kong) were firstly wet-cleaned with 5% detergent solution (Pyroneg) at 90 °C for 20 min. The substrates were immersed in deionized (DI) water and successively ultra-sonicated in DI water, acetone, and isopropanol for 10 min. Next, they were treated with UV-ozone irradiation for 20 min. The ZnO interface layer (25 nm) was spin-coated from a zinc oxide sol−gel precursor, which was prepared as described in previous studies [24]. ZnO films were annealed at 280 °C for 10 min.

### 2.3. Device Fabrication

For device fabrication in an oxygen- and moisture-free environment, thin ZnO-coated ITO-glass substrates were transferred into a glove box system filled with nitrogen. Polymer solar cells were fabricated using an inverted architecture; ITO-glass/ZnO/(PTNT:PC_71_BM)/MoO_3_/Ag. 

Photoactive layers were deposited from the non-halogenated *o*-xylene main solvent without an additive or together with non-chlorinated additives (MN and PN) having film thicknesses greater than 200 nm. DIO and CN were added to *o*-xylene as solvent additives as well to prepare the films. The BHJ of PTNT:PC_71_BM was spin-coated with 1000 rpm for 60 s, and dried with 3000 rpm for 30 s from a 2:3 donor:acceptor ratio in *o*-xylene (or other solvent blends (vol %)), as the processing solvent. Spin-coated films were allowed to dry for 3 h in the evaporator (<10^−6^ mbar) prior to the electrode deposition. A molybdenum oxide hole transport layer (MoO_3_) (12 nm) and metallic Ag (80 nm) contacts were deposited using a shadow mask by thermal evaporation (Covap system supplied by Angstrom Engineering) under 1 × 10^−6^ mbar inside the glove box. The active area of the solar cells was defined to 0.10 cm^2^ by a mask during the testing. The J–V data were measured using a Keithley 2400 Source Meter under AM 1.5 G illumination from an Oriel solar simulator with a Newport 150 W xenon lamp giving a light intensity output of 100 mV/cm^2^. 

### 2.4. Photo-Physical Properties

Photoluminescence (PL) measurements were conducted using a Cary Eclipse Fluorescence spectrophotometer (Santa Clara, CA, USA). The sample holder was positioned at a constant height with several angles of incidence of the excitation, which may be varied from 20°–35°. The PL intensities were normalized with respect to the thickness of the films as previously described [25].

### 2.5. Film Topography

The surface topography of the samples was analyzed using multimode atomic force microscopy (AFM) (supplied by Bruker, Billerica, MA, USA) in tapping mode using a J-scanner and Si tips. Samples were previously dried inside the glove box overnight.

## 3. Results and Discussion

### 3.1. Solvent and Solvent Additives

This study aimed to explore the use of a less-toxic non-halogenated solvent (*o*-xylene) and solvent additives (MN and PN) with selective solubility in the fabrication of organic solar cells. However, chlorinated solvent additives (CN and DIO) together with *o*-xylene were also used in order to make a comparison between the two types of solvents (non-chlorinated vs. chlorinated) and understand the potential of the non-chlorinated solvents in the fabrication of high-performing photovoltaic devices. It has been shown that some solvent additives can enhance the crystallinity and the available interfacial area between the donor and the acceptor in BHJs, increasing the charge carrier mobility and device efficiency [26].

We selected *o*-xylene and solvent additives that are different from those used in previous studies of PTNT solar cells. Chlorinated solvents are banned from industrial use in many countries and should be avoided if possible. Another reason for the use of the non-chlorinated solvents in the current study is due to their lower toxicity, which is an important criterion. The physical properties such as the boiling point and vapor pressure of the solvents are given in Table 1, with additives generally being characterized with a low vapor pressure and high boiling point, and hence leading to slow drying compared to the host solvent (*o*-xylene) during the film fabrication process. Note that additives dissolve the acceptor better than the polymer. Environmental and health impacts of the various solvents and solvent additives according to Chemwatch hazard ratings and statements are given in Table 2.

As shown in Table 2, the non-halogenated solvent additives (MN and PN) have slightly different hazard ratings compared to the halogenated ones (CN and DIO). However, the non-halogenated solvents have much fewer hazard statements, suggesting that they are less harmful and less toxic to the environment as compared to the halogenated solvents. Based on the safety data from Chemwatch, the toxicity of the additives is listed in the order of CN (2) > DIO (2) > PN (2) > MN (1). It is reported that MN has a chronic score of 2, as seen in Table 2, whereas the halogenated counterpart has that of 0 based on the current Chemwatch material safety data sheet (MSDS). From the data, it is clear that there are only small differences between the solvent additives and *o*-xylene is actually the most harmful solvent used in this study according to the Chemwatch ratings. Xylenes are widely used in different industrial applications, but even less harmful alternatives would be a big advantage for solar cell preparation. The use of water as the processing solvent would be a favorable alternative; however, using PTNT:PCBM nanoparticles dispersed in water gives a quite low performance in solar cells [29]. Recently a method of preparing high-performance solar cells from aqueous nanoparticles has been described [30]. 

### 3.2. Photovoltaic Properties

The inverted architecture (glass/ITO/ZnO/PTNT:PC_71_BM/MoO_3_/Ag) was selected as it has a higher stability than conventional devices due to the absence of a low work function electrode. ZnO is one of the most commonly used inorganic components in organic solar cells [31,32]. It has a lower LUMO energy level than that of the acceptor, which is helpful for the extraction of electrons, and it has a lower HOMO level than that of the polymer donor, and hence blocks the holes from flowing towards the ITO cathode. MoO_3_ (molybdenum trioxide) acts as a hole transport layer in inverted organic solar cells. It helps to prevent the diffusion of metal ions and oxygen into the active layer, and thus potentially provides a long life span [33]. In this study, MoO_3_ plays an important role as its HOMO level of −5.3 eV is close to that of PTNT (−5.0 eV), enabling the effective extraction of holes from the active layer which were determined by ultraviolet photoelectron spectroscopy studies in previous work [11]. The electrical properties at the organic–electrode interfaces also affect solar cell performance greatly as the series resistance (RS) of a solar cell is determined by both the electrical resistivity of each layer and the contact resistance between the layers. Here, the LUMO level of MoO_3_ is much higher than that of PC_71_BM, which also greatly reduces the recombination of charge carriers at the organic–MoO_3_ electrode interface. Surface energies can have a significant impact in altering the phase separation processes during the coating and deposition of the layers, as well as the drying and annealing of the BHJ blend film. The buffer interfacial layers also need to adhere to the surface well enough, yet the work function values need to be kept the same. It is reported that the performance of the polymer solar cell can be enhanced with such energy control treatment [34,35].

For the non-halogenated solvent system for the fabrication of the inverted BHJ devices, *o*-xylene was used as it is less harmful than chlorinated solvents and dissolves both the donor polymer (PTNT) and PC_71_BM very well. Moreover, MN and PN were used as solvent additives as they are less hazardous and PN is known to enhance the crystallinity and the interfacial area between the donor and the acceptor in most BHJs [36]. The photo-active films of PTNT:PC_71_BM were first optimized for the best outcomes based on the donor–acceptor ratio and spin-coating conditions. Later, solvent additives were used to optimize the nanomorphology. 

Devices prepared from a 2:3 PTNT:PC_71_BM (wt/wt) blend using *o*-xylene as a processing solvent without any additive yielded a PCE of 1.5% with a low J_SC_ of 2.5 mAcm^−2^, due to the non-optimized film morphology. A ratio of 1:2 PTNT:PC_71_BM (wt/wt) blend showed even lower efficiencies. Both non-halogenated solvent additives (MN, PN) and halogenated additives (DIO, CN) in *o*-xylene were used to improve the film quality and generally resulted in improved photovoltaic devices as summarized in Table 3. 

Scratched AFM images were used to estimate the film thickness. The photoactive layer with the PN solvent additive was ~250 nm and the other films were around 210–230 nm. Among the solvent additives, 2% MN yielded a maximum PCE of 6.2% with a corresponding Voc = 0.88 V, J_SC_ = 10.0 mAcm^−2^, and FF = 70% (Figure 2a and Table 3). This is the highest performance reported for the high band gap PTNT polymer to date using an entirely non-halogenated solvent system. In addition, the performance is higher than the PCE reported using *o*-DCB as a solvent [7]. Inspired by the promising results achieved from *o*-xylene and MN, 3% of the PN solvent additive on *o*-xylene was also investigated and gave a promising PCE of 5.4% with a corresponding J_SC_ of 10.4 mAcm^−2^ and a V_OC_ of 0.837 V. The slightly lower performance achieved based on the PN additive as compared to MN is mainly due to the lower FF (59%) of the devices, which could be attributed to its high boiling point (Table 1), hence affecting the morphology of the film. DIO and CN solvent additives were used for comparison with the non-halogenated solvent additives and gave a 5.2% and 3.9% PCE, respectively, with both offering lower performances as compared to the greener solvent additives (MN and PN). The amounts of the solvent additives used are the optimal concentrations found in this study after using a few different solvent and PTNT:PC_71_BM ratios. Figure 2b shows the external quantum efficiency (EQE) of the devices prepared with the addition of the various solvent additives. Calculated theoretical currents from the EQE data can be found in the Appendix A and reveal a slight overestimation of the measured currents. The use of MN and PN solvent additives significantly improved the photon-to-electron conversion efficiency with the response reaching more than 70% in the wavelength range of 400 to 550 nm, which is in agreement with the higher J_SC_ obtained as seen in Figure 2a and Table 3. The use of the halogenated solvent additives (DIO and CN) gave a lower response while the device without solvent additive was inferior in terms of the response.

### 3.3. Photoluminescence (PL) 

In order to understand the role of the solvent additives on the efficiency of the exciton dissociation, the PL emission of the BHJ blends were measured and are shown in Figure 3.

The PL emissions of the films processed with and without solvent additives were compared. The PL spectrum from PTNT:PC_71_BM without using any additive is clearly observed which could be due to the larger phase segregation and poor charge transfer from the donor to the acceptor phase, which is consistent with the lower J_SC_ of the photovoltaic devices. The emissions from the blends drastically dropped, with an extensive energy transfer with the use of the solvent additives. The blend with the CN solvent additive still had some emission at around 600 nm suggesting that there was still a reduced quenching of the excited states and hence the blend also showed a lower device performance. The PL of the photovoltaic devices after the use of MN, PN, and DIO was significantly quenched, consistent with the higher currents measured and consequently resulting in a higher PCE. 

### 3.4. Surface Topography

AFM was employed to gain a deeper understanding of the effect of the various solvent additives on the surface topography of the blend films, as shown in Figure 4. The photoactive films of the polymer solar cells clearly had a much smoother surface when additives were used. Additives can be categorized according to their effect as either improving the phase separation or suppressing the oversized phase segregation [12,27]. In our study, they enhanced the intermixing of the materials to increase the donor–acceptor interface, allowing the efficient dissociation of excitons into holes and electrons.

Different solvent additives have been used to process the photoactive layer, as reported in earlier studies. The purpose here was to identify whether less harmful additives would tune the nanomorphology better than their harmful counterparts. Using non-halogenated MN and PN, the intermixing of donor–acceptor moieties was immensely enhanced, promoting the efficient charge transport as investigated in the AFM study. It is important to note, however, that the inner morphology of the BHJ devices can be different as AFM is a surface-sensitive technique [37,38].

As seen from the AFM images in Figure 4, the PTNT:PC_71_BM film deposited from *o*-xylene appeared extremely phase separated and the surface was rough with a root mean square (rms) of 12 nm. Nevertheless, there was a visible difference in surface topography of the films showing better intermixing when solvent additives of CN, DIO, PN, and MN were employed in various volume percentages. The smooth morphology and intermixing observed is likely to improve the charge separation and transport, hence improving the J_SC_ and FF of the photovoltaic devices. It was also noted that the films formed without using any additive were cloudy, while they needed to be clear enough for light harvesting. The AFM images were in good agreement with the photovoltaic device performances (Table 3). The roughness of the topographies differed slightly, as given in Table 4. The MN-processed active layer had the lowest rms value which could indicate a better intermixing and finer interpenetrating network across the film for the best charge transport. Thus, the higher PCE (6.0% > 5.2%) compared to the PN-processed active layer could also be attributed to the lower roughness of the film (2.03 nm < 2.66 nm). Moreover, the use of MN as a solvent additive might have led to a well-organized bulk morphology with a lower content of PC_71_BM embedded in the blend matrix. Consequently, the non-halogenated and less-toxic MN gave the highest PCE (6%). 

In brief, the high device efficiency generated by using the greener processing solvent is due to the enhanced morphology of the photoactive layer, enabling the extraction of a relatively high J_SC_ and FF. 

## 4. Conclusions

The preparation of environmentally friendlier polymer solar cell devices was achieved and their characterization was done successfully, revealing their photovoltaic performances. PTNT-conjugated polymer was used as the donor polymer and non-halogenated *o*-xylene was successfully used as the main solvent together with two non-chlorinated solvent additives. The best device performance was achieved with MN as the solvent additive with an average PCE of 6%. The morphology and device studies revealed that the additives improved the nanostructures of donor–acceptor blends and increased the short-circuit currents. The successful use of the greener processing solvents for the fabrication of high-performing polymer solar cells is a crucial step forward towards the ultimate future goal of producing solar cells on flexible substrates and on a large scale.

## Figures and Tables

**Figure 1 polymers-11-00544-f001:**
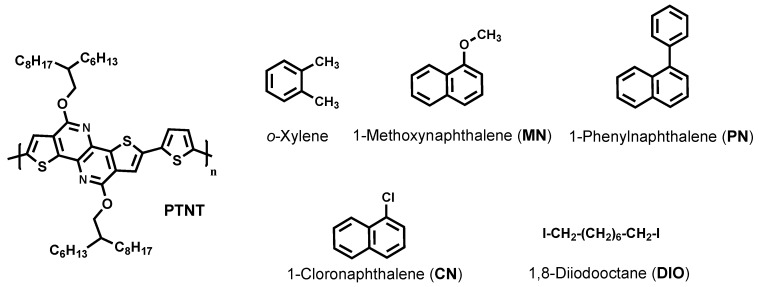
Chemical structures of poly (2,5-thiophene-alt-4,9-bis(2-hexyldecyl)-4,9-dihydrodithieno[3,2-c:3′,2′h][1,5] naphthyridine-5,10-dione (PTNT), *o*-xylene, and solvent additives used in the bulk heterojunction (BHJ) solar cell.

**Figure 2 polymers-11-00544-f002:**
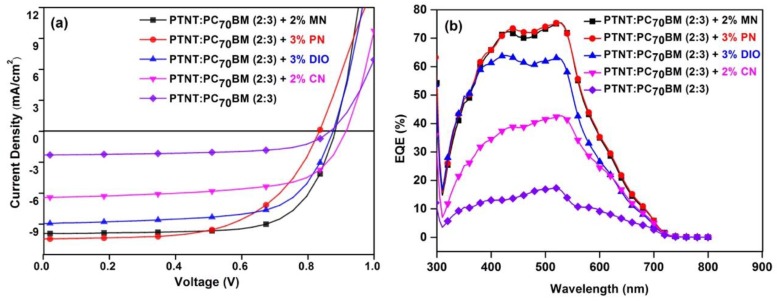
Photovoltaic devices with and without solvent additives: (**a**) representative current density–voltage curves; (**b**) EQE versus wavelength.

**Figure 3 polymers-11-00544-f003:**
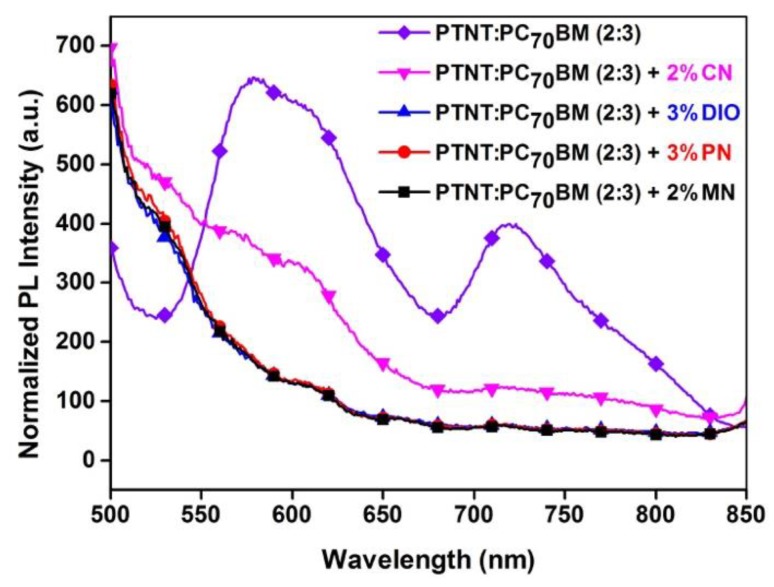
Normalized photoluminescence (PL) emission spectra of active layers from PTNT:PC_71_BM blends using different solvent additives (normalized to film thickness).

**Figure 4 polymers-11-00544-f004:**
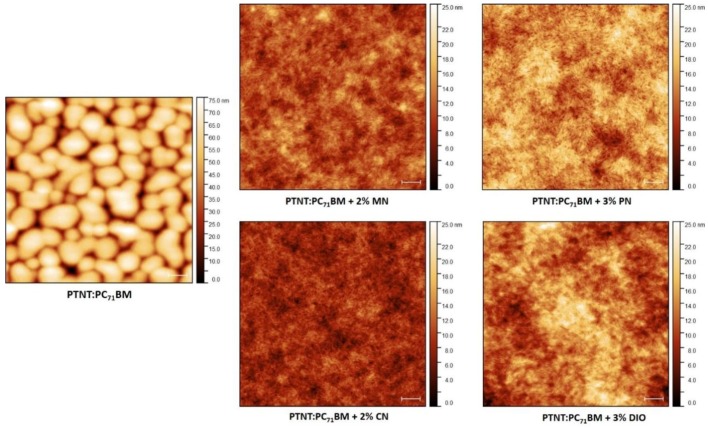
Atomic force microscopy (AFM) images (5 µm × 5 µm) showing the surface topography of PTNT:PC_71_BM (2:3) blend from *o*-xylene with different solvent additives at room temperature. The scale bar is 500 nm.

**Table 1 polymers-11-00544-t001:** Various physical properties of solvents and solvent additives used in this work [27].

Solvent	Molecular Formula	Boiling Point (°C)	Vapor Pressure (kPa at 25 °C)
*o*-Xylene	C_8_H_10_	144	0.881
1-Methoxynaphthalene (MN)	C_11_H_10_O	270	0.002
1-Phenylnaphthalene (PN)	C_16_H_12_	324	0.0003
1-Chloronaphthalene (CN)	C_10_H_7_Cl	259	0.003
1,8-Diiodooctane (DIO)	C_8_H_16_I_2_	333	0.00004

**Table 2 polymers-11-00544-t002:** Hazard identification and classification of the solvents [28].

Solvent	Flammability	Toxicity	Body Contact	Reactivity	Chronic	Hazard Alert Code
*o*-xylene	2	2	2	1	0	2
1-methoxynaphthalene (MN)	1	1	1	1	2	2
1-phenylnaphthalene (PN)	1	2	1	1	0	2
1-chloronaphthalene (CN)	1	2	2	1	0	2
1,8-diiodooctane (DIO)	1	2	2	1	0	2

(0 = minimum, 1 = low, 2 = moderate, 3 = high, 4 = extreme).

**Table 3 polymers-11-00544-t003:** Summary of the photovoltaic performance properties of optimized PTNT:PC_71_BM (2:3) solar cells with mean values and standard deviations from six devices.

Device Structure ITO/ZnO/BHJ/MoO_3_/Ag		Solvent + (*v*/*v*) Additive	J_sc_ (mA cm^−2^)	Voc (V)	FF (%)	PCE (%)	
						Mean	Max
		*o*-xylene	2.5 ± 0.1	0.877 ± 0.016	62 ± 1	1.4 ± 0.1	1.5
		*o*-xylene + 2% MN	9.9 ± 0.2	0.879 ± 0.007	69 ± 1	6.0 ± 0.1	6.2
PTNT:PC_71_BM (2:3) 1000 rpm		*o*-xylene + 3% PN	10.4 ± 0.4	0.837 ± 0.003	59 ± 1	5.2 ± 0.2	5.4
		*o*-xylene + 3% DIO	8.6 ± 0.6	0.873 ± 0.002	65 ± 1	4.9 ± 0.4	5.2
		*o*-xylene + 2% CN	6.5 ± 0.2	0.917 ± 0.008	63 ± 1	3.7 ± 0.1	3.9

**Table 4 polymers-11-00544-t004:** Root mean square (rms) values of BHJ films treated with various solvent additives.

Additives	Without Additive	2% MN	3% PN	2% CN	3% DIO
rms (nm)	12.9	2.03	2.66	1.81	3.20

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
