# Peer review of "Optimizing Polymer Solar Cells Using Non-Halogenated Solvent Blends"

_polymers, 2019, doi:10.3390/polym11030544_

Round 1

Reviewer 1 Report

In this manuscript, the authors optimized polymer solar cells using non-halogenated solvent blends. They show the less harmful system gave a 20% rise in PCE as compared to an earlier reported device efficiency that was processed from a chlorinated solvent (o-DCB). However, there are still some issues which should be addressed.

(1)   How about the integrated current from the IPCE results? Is there deviation from the I-V measurements?

(2)   How does the content of the additives affect the morphology and the device performance? Is the present content optimal?

(3)   Is there are more experimental measurements results about the film morphology?

(4)   In line 233, Figure 9 should be Figure 4.

(5)   How about the device stability?

(6)   Some recent results should be included in this manuscript. Polymers 2018, 10, 127; doi:10.3390/polym10020127; Polymers 2018, 10(7), 725; https://doi.org/10.3390/polym10070725

Author Response

    Reviewer 1.

    In this manuscript, the authors optimized polymer solar cells using non-halogenated solvent blends. They show the less harmful system gave a 20% rise in PCE as compared to an earlier reported device efficiency that was processed from a chlorinated solvent (o-DCB). However, there are still some issues, which should be addressed.

(1)         How about the integrated current from the IPCE results? Is there deviation from the I-V measurements?

Response: Thank you for the comment. Yes, we have calculated the theoretical integrated current from the IPCE and compared it with the experimental short circuit current from the I-V measurements, which a minor overestimation in the experimental is observed. However, it important to note that the role of the additives played in optimizing/ improving the device efficiencies compared to the ones fabricated without additives.

    (2)   How does the content of the additives affect the morphology and the device performance? Is the present content optimal?

Response: The use of additives (both chlorinated and non-chlorinated solvents) generally have resulted in an improved device efficiency mainly due to the fine-tuning of the surface morphology of the blend film. For instance, the devices processed from 2% methoxynaphthalene (MN) in o-xylene (v/v) gave over 6% power conversion efficiency, compared to a mere 1.5% power conversion efficiency of the device processed with o-xylene only in the absence of MN solvent additive. Even though, the main focus of the study was to explore the potential of non-chlorinated solvents, we have also fabricated devices using chlorinated solvent additives in order to make a fair comparison. Interestingly, the devices fabricated from non-chlorinated solvents gave a higher efficiency compared to the commonly used chlorinated solvent additives, showing the high potential and bright prospect of non-chlorinated and green solvent systems towards fabricating high performing photovoltaic devices. However, we have not looked into the effect of different amount of solvent additive loading on the morphology evolution of the devices due to the limited scope of the study. However, the suggestion is well received and will be considered in a separate study in the future.

The role of solvent additives on the morphology of bulk heterojunction films was one of the topics that have been investigated in the past. The solvent additives usually have a high boiling point (dry slowly during film making) and have higher tendency to dissolve the acceptor (PC71BM) than the polymer. Therefore during the film making process (drying process), the acceptor will dry slowly which will give a good opportunity for the polymer chains to semi-crystallize and organize fairly well. The process would encourage towards the formation of two pure phases that will facilitate the facile exciton dissociation and transport of charge carriers (https://www.sciencedirect.com/science/article/pii/S1369702113002915). Note that the amount and nature of the additive needs to be controlled properly in order to find a good balance between phase separation and inter mixing of the donor and acceptor, which both are important for the charge dissociation and transport of charge carriers to ultimately achieve a high device efficiency.

(2)         Is there are more experimental measurements results about the film morphology?

Response: We have only looked the morphology using AFM.

    (4)   In line 233, Figure 9 should be Figure 4.

Response: We have made this change in the revised manuscript.

    (5)   How about the device stability?

Response: We totally agree with the reviewer on the importance of device stability on the real life application of polymer solar cells. However, the scope of the current study was limited to investigating the role of green and non-chlorinated processing solvents on device efficiency.  We would however like to mention that a study on device stability will be initiated within our group in a completely new study in the near future.

    (6)   Some recent results should be included in this manuscript. Polymers 2018, 10, 127; doi:10.3390/polym10020127; Polymers 2018, 10(7), 725; https://doi.org/10.3390/polym10070725

  Response: The suggested articles are cited in the revised manuscript.

Reviewer 2 Report

The authors are involved in the morphology optimization of the photoactive layer of PTNT-PC71BM inverted BHJ solar cells by using some environmentally friendly organic solvents. The study is well presented and the conclusion are adequately supported. However, some points need some improvements.

1) line 83 Which kind of column (and detector) has been used for SEC characterization?

2) Line 91 What is pyroneg?

3) Line 115: Why "simple" readings?

4) Line 127 Fullerene molecules are pre-dispositioned.... better explain this sentence

5) Lines 156 and 161: MoO3 the number 3 must be subscript

6) Line 176 The toxicity list can be omitted (since it already compares at line 141)

7) Table 3: The additive percentage is expressed in v/v?

8) Figure 3. The size of the curves markers should be expanded

9) Lines 277-279 These sentences should be better explained

10) Introduction. Authors should include a short reference to the use of water-soluble polythiophenes for green BHJ solar cells (doi: 10.1016/j.polymer.2018.07.012)

Author Response

Reviewer 2

    The authors are involved in the morphology optimization of the photoactive layer of PTNT-PC71BM inverted BHJ solar cells by using some environmentally friendly organic solvents. The study is well presented and the conclusion are adequately supported. However, some points need some improvements.

    1) line 83 Which kind of column (and detector) has been used for SEC characterization?

Response: The column and detector type used for the polymer characterization is mentioned in the materials and methods section of the revised manuscript.

    2) Line 91 What is pyroneg?

Response: Pyroneg is a glass cleaning detergent/ chemical that has been used for cleaning the ITO glass substrates, which is clarified in the manuscript.

    3) Line 115: Why "simple" readings?

Response: The word ‘simple’ in the manuscript was referring to the reading from the PL spectrometer before the correction is made taking the thickness of the films in to consideration. We have rewritten the sentence and avoided the word ‘simple’ in the revised manuscript.

    4) Line 127 Fullerene molecules are pre-dispositioned.... better explain this sentence

Response: The sentence was rewritten and explained in a better way.

    5) Lines 156 and 161: MoO3 the number 3 must be subscript

Response: We have corrected the formula.

    6) Line 176 The toxicity list can be omitted (since it already compares at line 141)

Response: We have deleted the repeated toxicity list.

    7) Table 3: The additive percentage is expressed in v/v?

Response: That is right and we have included the unit (v/v) in Table 3.

    8) Figure 3. The size of the curves markers should be expanded

    Response: As suggested, the size of the curve markers in Figure 3 were expanded.

    9) Lines 277-279 These sentences should be better explained

    Response:  The sentences were rewritten and explained in a better way.

    10) Introduction. Authors should include a short reference to the use of water-soluble polythiophenes for green BHJ solar cells (doi: 10.1016/j.polymer.2018.07.012)

    Response: The suggested article was cited in the revised manuscript.

Reviewer 3 Report

The authors present polymer solar cells consisting of PTNT and PC71BM processed out of o-xylene solution. The influence of a few halogenated and non-halogenated solvent additives is investigated. Some aspects have to be improved in my opinion:

In line 21 "BJH films of the best devices were optimized to achieve an ideal morphology" is hard to understand. The term "best devices" already implies some optimization process.

The abstract contains abbreviations which are not explained: BJH, o-DCB.

In the abstract and throughout the text some of the chemicals and equipment are capitalized (Atomic Force Microscopy, Glove Box).

Abbreviations should be used consistently (1-MN vs MN). It is ok to use the abbreviation MN for 1-methoxynaphtalene throughout the text. On the other hand, "o-DCB" in line 37 is not the matching abbreviation for "dichlorobenzene".

For one of the solvent additives Nerolin Bromelia, shown in fig. 1, mentioned in the materials section (line 86) and device fabrication (as "BN" in line 101) no results are shown or discussed.

The aspect of a "green" solvent system is overly emphasized. I do not see the specific order of toxicity given in line 141 based on a nontransparent rating system of a company (chemwatch). Furthermore in the caption of table 2 these different ratings (0 - 4) are wrongly linked to the GHS system. What is the point to discuss small differences in toxicity/hazardousness of solvent additives, if the main solvent is rated worse than any of the additives?

In line 168 it is said "solar cells were prepared from non-halogenated solvent systems" which is not the case for all the solar cells discussed.

In line 184 Supplementary is referenced. Does this exist?

The graphs in figure 2 and 3 should use the same color code. Image quality is bad, likely a problem with conversion of image file formats.

Line 229 "lower-toxicity which depends on solvent polarity". What does this mean?

The morphology is solely discussed in terms of surface morphology obtained by AFM. It should at least be mentioned that the inner morphology of the BHJ decisive for solar cell performance can be quite different. (Adv. Funct. Mater. 21, 3382-3391, 2011; Adv. Funct. Mater. 28, 1800209, 2018).

Author Response

    Reviewer 3

    The authors present polymer solar cells consisting of PTNT and PC71BM processed out of o-xylene solution. The influence of a few halogenated and non-halogenated solvent additives is investigated. Some aspects have to be improved in my opinion:

    In line 21 "BJH films of the best devices were optimized to achieve an ideal morphology" is hard to understand. The term "best devices" already implies some optimization process.

       Response:  The term “best devices” was modified for clarity in the revised manuscript.

    The abstract contains abbreviations which are not explained: BJH, o-DCB.

    Response: The abbreviations are written in full.

    In the abstract and throughout the text some of the chemicals and equipment are capitalized (Atomic Force Microscopy, Glove Box).

Response: We have gone through the manuscript and the required changes were made.

    Abbreviations should be used consistently (1-MN vs MN). It is ok to use the abbreviation MN for 1-methoxynaphtalene throughout the text. On the other hand, "o-DCB" in line 37 is not the matching abbreviation for "dichlorobenzene".

   Response: The name has been modified to o- dichlorobenzene and also MN was used throughout the text in the revised version.

    For one of the solvent additives Nerolin Bromelia, shown in fig. 1, mentioned in the materials section (line 86) and device fabrication (as "BN" in line 101) no results are shown or discussed.

Response: We have removed Nerolin Bromelia from the text as the study was only focusing on the following solvent additives: MN, PN, DIO and CN.

    The aspect of a "green" solvent system is overly emphasized. I do not see the specific order of toxicity given in line 141 based on a nontransparent rating system of a company (chemwatch). Furthermore in the caption of table 2 these different ratings (0 - 4) are wrongly linked to the GHS system. What is the point to discuss small differences in toxicity/hazardousness of solvent additives, if the main solvent is rated worse than any of the additives?

Response: Yes, the hazard ratings are similar. However, there is a clear difference in the hazard statements between the halogenated and non-halogenated solvents in Chemwatch, revealing the negative impact of the former in the environment and human health. Moreover, it is widely agreed that halogenated solvents in general are a concern to human health and to the environment and hence their use are highly restricted/banned in industries. Therefore, if the polymer solar cell is to be viable for large scale application and get its market share in the future, we believe that the move towards the direction of using less harmful solvents is very important, which is why we initiated this project and explored the potential of the non-halogenated solvents in the fabrication of high performing devices. Energy Environ. Sci., 2015, 8, 585-591; Pure and Applied Chemistry 72, 1207–1228

    In line 168 it is said "solar cells were prepared from non-halogenated solvent systems" which is not the case for all the solar cells discussed.

    Response: Yes, chlorinated solvents have been also used in order to make a comparison with the non-chlorinated solvents and the sentence is rephrased.

    In line 184 Supplementary is referenced. Does this exist?

    Response: We removed the specific text regarding the Supplementary reference in that line.

We have added a supplementary information about the integrated currents from IPCE.   

    The graphs in figure 2 and 3 should use the same color code. Image quality is bad, likely a problem with conversion of image file formats.

    Response: We have drawn the graphs to improve the image quality and used the same color code.

    Line 229 "lower-toxicity which depends on solvent polarity". What does this mean?

    Response: We have modified the sentence to remove the non-obvious relationship between toxicity and solvent polarity.

    The morphology is solely discussed in terms of surface morphology obtained by AFM. It should at least be mentioned that the inner morphology of the BHJ decisive for solar cell performance can be quite different. (Adv. Funct. Mater. 21, 3382-3391, 2011; Adv. Funct. Mater. 28, 1800209, 2018).

    Response: We have added the suggested statement in the revised manuscript and the references are cited.
